# Peer review of "Investigation of SiC Trench MOSFETs’ Reliability under Short-Circuit Conditions"

_materials, 2022, doi:10.3390/ma15020598_

Round 1

Reviewer 1 Report

The work is on investigations of robustness of commercial SiC based trench MOSFET under short circuit conditions. The failure of the devices were tested and mode of failure is reported. Moreover, the devices were simluated to understand the exact behaviour. The work looks interesting to scientic community working in this area. The work may be accepted after inclusion of following comments:

  1. Line 24.. “manufacturers are available ……...” please provide references.
  2. Line 28, 29… How reliability is affected by said parameters is unclear. It would be better to provide some more discussion on the issues.
  3. Line 53 through 56.. To me, the structure of said “cell” won’t give same structure if repeated. How it is cell structure? Which fitting is used, how doping concentration was determined is unclear. Please mention the doping concentration and vertical dimensions of the devices.
  4. Table 4.. Why different parameters are used for both devices. How can you compare device characteristics based on different device parameters?
  5. Figure 9.. The meshing is not looking good to reviewer. A bad meshing causes misleading results.
  6. The authors are encouraged to improve the introduction section by inclusion of names of some other devices based on said properties. For example, following references may be cited:

Vibhor Kumar, Jyoti Verma, A.S.Maan, Jamil Akhtar "Epitaxial 4H–SiC based Schottky diode temperature sensors in ultra-low current range" vol 182, 109590, 2020. https://doi.org/10.1016/j.vacuum.2020.109590

SIQI LI, SIZHAO LU, AND CHUNTING CHRIS MI “Revolution of Electric Vehicle Charging Technologies Accelerated by Wide Bandgap Devices”  10.1109/JPROC.2021.3071977

Vibhor Kumar, A. S. Maan, Jamil Akhtar, "Barrier height inhomogeneities induced anomaly in thermal sensitivity of Ni/4H-SiC Schottky diode temperature sensor" Journal of Vacuum Science & Technology B 32, 041203 (2014); https://doi.org/10.1116/1.4884756

Author Response

Thank you very much for your kindly comments on our manuscript. There is no doubt that these comments are valuable and very helpful for revising and improving our manuscript. In what follows, we would like to answer the questions you mentioned.

Reviewer 2 Report

A review of the article is presented in the attached file.

Author Response

Thank you very much for you appreciation.

Reviewer 3 Report

The paper is very interesting and it deserves to be published. Howerver, the authors have to comment in the introduction on the possible contribution of crystalline defects in the semiconductor (i.e. threading dislocations) on the carrier generation inducing gate reliability issues

Author Response

Thank you for your suggestion. We have added the discussion of tthe threading dislocations in introduction part.

Reviewer 4 Report

Please see attached PDF file.

Author Response

Thank you for your positive and constructive comments and suggestions on our manuscript,and the specific response can be found in the attached document.

Round 2

Reviewer 1 Report

The authors have responded satisfactory on the reviewer's comments. The work may be consider for publication.